# Financial vulnerability and the impact of COVID-19 on American households

**Carol Bruce☉, Maeve E. Gearing[ID]\*☉, Jill DeMatteis‡, Kerry Levin‡, Timothy Mulcahy‡, Jocelyn Newsome‡, Jonathan Wivagg‡**

Westat, Rockville, Maryland, United States of America

☉ These authors contributed equally to this work.
‡ JD, KL, TM, JN and JW also contributed equally to this work.
\* MaeveGearing@westat.com

**Data Availability Statement:** The data files are housed at https://covid19.richdataservices.com/westat/westat_covid_2020/.

## Abstract

In May 2020, Westat, in partnership with Stanford University School of Medicine, conducted a nationally-representative household survey of American attitudes and behaviors regarding COVID-19. In this article, we examine what the Coronavirus Attitudes and Behaviors Survey tells us about the impact of COVID-19 on financial status and how this impact varies by demographic characteristics, the presence of health risk factors, and financial status (including employment factors). The survey reveals significant inequality in financial impact, as those who were most financially vulnerable prior to the pandemic found themselves under greater financial strain, while those who were more financially secure have experienced a neutral or even positive impact of the pandemic on household finances. These findings have important implications for public policy as policymakers seek to target aid to those who need it most.

## Introduction

The 2020 pandemic of novel coronavirus disease (COVID-19) has had a pervasive impact on almost all aspects of life, including the ability of individuals and households to financially support themselves. During February and March 2020, the total number of hours worked fell by 60 percent, driven by layoffs from employment and closure of businesses [1]. Hours worked increased in mid-April 2020, leveling off at 25 percent below the January 2020 baseline, and remaining at 35 to 30 percent below baseline through August 2020 [2].

Lower-income households with children experienced loss of income disproportionately [3–7]. The U.S. Census Bureau's Household Pulse Survey in July 2020 showed that more than 60 percent of low-income households with children experienced an income shock due to COVID-19, resulting in food insecurity and difficulty paying bills [2, 3]. Food insecurity in households with children rose in July 2020 to 32 percent, more than double rates measured in 2018 [2]. Larger households are most vulnerable to falling behind on household finances due to COVID-19, with each child in a household increasing the likelihood of financial delinquency by an additional 17 percent [5].

Hispanic and Black households also report disproportionate loss of income from the pandemic. Approximately 70 percent of Hispanic household heads reported experiencing an

**Funding:** Westat provided funding in the form of salaries for CB, MG, JDM, KL, TM, JN, and JW. The specific roles of these authors are articulated in the 'author contributions' section. The funders had no role in study design, data collection and analysis, decision to publish, or preparation of the manuscript.

**Competing interests:** The authors have read the journal's policy and have the following competing interests: All authors are paid employees of Westat, a private research company. This does not alter our adherence to PLOS ONE policies on sharing data and materials. There are no patents, products in development or marketed products associated with this research to declare.

income shock during the pandemic, as of July 2020, compared to 60 percent of Black households and 50 percent of non-Hispanic White households [2]. Reductions in consumption following the income shock were 50 percent higher in Black households, and 20 percent higher in Hispanic households, compared to non-Hispanic White households [4]. The Pew Research Center's American Trends Panel finds similar patterns in racial disproportionality, with 61 percent of Hispanics and 44 percent of Blacks reporting a loss of employment and wages during the coronavirus outbreak, compared to 38 percent of White respondents. Economic hardship is also reported at higher rates for racial and ethnic minorities, with 73 percent of Black and 70 percent of Hispanic adults reporting that they lacked emergency funds to cover three months of expenses (compared to 47% of Whites), and Blacks and Hispanics more often reporting that they would be unable to fully pay bills in April 2020 (48 percent Black, 44 percent Hispanic, vs 26 percent White respondents [6].

Households have responded to the loss of income and the threat of loss of income with a sharp decline in aggregate spending and an associated increase in saving since the onset of the pandemic [8–11]. Households' ability to save was enhanced by the provision of government-funded stimulus payments and increased unemployment benefits [11–15]. However, the ability to reduce household spending and increase household saving is not universal. Recent household surveys reveal that one in five households have depleted their savings, fallen behind on housing payments, or are experiencing difficulty paying debts, buying groceries, and paying utilities [6, 7, 16–18].

This study will expand on this research regarding the impact of the COVID-19 pandemic on household finances. Specifically, the analysis examines whether those who are most vulnerable financially are disproportionately experiencing a negative financial impact from the pandemic. We also explore demographic variation in the impact of the pandemic on household finances, and how the financial impact varies by employment factors and for individuals with health risk factors.

## Methods

### Sample description

This study was approved by the Westat Institutional Review Board. The project number is 1065, FWA 00005551. Informed consent was obtained in writing.

The Coronavirus Attitudes and Beliefs survey was administered in May 2020. An invitation to participate in the online survey was mailed to 13, 590 randomly selected households across the United States. The online survey included a discussion of the risks and benefits of participation and solicited informed consent. The survey included questions regarding individuals' beliefs about risk of becoming infected, likely consequences of contracting the virus, concerns about the impact of the virus, impact of the pandemic on household finances, and changes in behavior since the onset of the pandemic. After subtracting surveys that were not deliverable, 10 percent of those invited to participate completed the survey, yielding a sample of 1,222 respondents. The sample was weighted and adjusted for nonresponse, to yield a nationally representative sample for analysis. While low, the response rate is in line with that found in other nationally-representative web surveys [19]. However, we recognize the need for caution in the interpretation of these descriptive results; we include 95 percent confidence intervals, adjusted for non-response, with all findings.

Table 1 provides the distribution of the analysis sample by age, gender, race and ethnicity, marital status, size of household, education, and household gross annual income. The sample aligns with the distribution of the U.S. population on nearly all factors, with a slight disproportionality by race and by education. Black adults are underrepresented (9 percent in the sample

**Table 1. Demographics characteristics.**

|  | Sample Size | Weighted Frequency | Standard Error | Percentage |
|---|---|---|---|---|
| **Total** | 1222 | 253815197 | 3324800 | 100.00 |
| **Age** |  |  |  |  |
| 18 to 25 yrs | 113 | 40461384 | 2401859 | 16.69 |
| 26 to 45 yrs | 362 | 76425696 | 3235669 | 31.52 |
| 46 to 65 yrs | 416 | 76962446 | 3219014 | 31.74 |
| 65 yrs or older | 301 | 48630038 | 2041588 | 20.10 |
| **Gender** |  |  |  |  |
| Male | 531 | 122565576 | 626686 | 48.72 |
| Female | 681 | 129019577 | 765188 | 51.28 |
| **Race/Ethnicity** |  |  |  |  |
| Hispanic | 94 | 40874758 | 259877 | 16.33 |
| White | 1020 | 189238372 | 2145663 | 74.56 |
| Black | 67 | 22697827 | 2265332 | 8.94 |
| Asian | 76 | 20798513 | 2378938 | 8.19 |
| American Indian/ Alaskan Native | 20 | 8014481 | 2234061 | 3.16 |
| Native Hawaiian/ Pacific Islander | 10 | 2144684 | 970601 | 0.85 |
| Other | 44 | 13374093 | 2060264 | 5.27 |
| **Marital Status** |  |  |  |  |
| Married | 664 | 131701352 | 4646478 | 52.15 |
| Widowed | 63 | 9943835 | 1574697 | 3.94 |
| Divorced | 162 | 26580270 | 2975455 | 10.53 |
| Separated | 23 | 5428520 | 1299325 | 2.15 |
| Never married | 302 | 78873602 | 4042989 | 31.23 |
| **Number in Household** |  |  |  |  |
| 1 | 236 | 31276237 | 2782316 | 12.51 |
| 2 | 465 | 85807109 | 4127375 | 34.32 |
| 3 | 207 | 49197103 | 3799333 | 19.68 |
| 4 | 177 | 44861293 | 4230072 | 17.94 |
| 5 or more | 122 | 38862849 | 8448049 | 15.55 |
| **Education** |  |  |  |  |
| Less than HS diploma | 39 | 29883311 | 1063300 | 11.86 |
| HS graduate or GED | 166 | 69172779 | 624789 | 27.44 |
| Some college or technical school | 200 | 51349468 | 2510793 | 20.37 |
| Associate's degree or professional certificate | 117 | 26507343 | 2510793 | 10.52 |
| Bachelor's degree | 399 | 46273239 | 1757095 | 18.36 |
| Master's or doctorate | 293 | 28860538 | 1903888 | 11.45 |
| **Annual Gross Household Income** |  |  |  |  |
| Less than 25k | 118 | 35434915 | 3961825 | 17.90 |
| 25k – 49,999 | 175 | 43781293 | 3643766 | 22.11 |
| 50k - 74,999 | 197 | 37463241 | 3701604 | 18.92 |
| 75k – 149,999 | 299 | 52579998 | 4033776 | 26.55 |
| 150k or more | 210 | 28758236 | 2788169 | 14.52 |

Source: Coronavirus Attitudes and Behaviors Survey, 2020.

versus 13 percent nationally), while Asians are overrepresented (8 percent in the sample versus 5 percent nationally). Regarding education, those who have some college or technical school are overrepresented (31 percent in the sample versus 26 percent nationally), while those who

**Table 2. Health factors.**

| | Sample Size | Weighted Frequency | Standard Error | Percentage |
|---|---|---|---|---|
| **Total** | 1222 | 253815197 | 3324800 | 100.00 |
| **Health Risk Factors** | | | | |
| None | 509 | 111990349 | 4942789 | 47.55 |
| 1 | 346 | 67684006 | 4777217 | 28.74 |
| 2 | 166 | 36180999 | 3501416 | 15.36 |
| 3 or more | 101 | 19650563 | 4637792 | 8.34 |
| **Diagnosed Conditions** | | | | |
| High blood pressure | 355 | 73697114 | 4103066 | 29.95 |
| Depression/Anxiety | 314 | 61285902 | 3987800 | 25.18 |
| Respiratory condition | 173 | 38819061 | 3479973 | 15.99 |
| Diabetes | 108 | 21950878 | 2681388 | 9.01 |
| Heart disease | 96 | 19292446 | 2816373 | 7.91 |
| Autoimmune disorder | 89 | 12527409 | 1678676 | 5.18 |
| Kidney disease | 28 | 4986774 | 1468621 | 2.07 |
| **COVID-19 Symptoms** | | | | |
| Yes | 157 | 27828511 | 2946696 | 11.03 |
| No | 1057 | 224387138 | 3011674 | 88.97 |
| **Tested for COVID-19** | | | | |
| Yes | 59 | 11911173 | 2056822 | 4.71 |
| No | 1160 | 241071511 | 2037478 | 95.29 |
| **Tested Positive for COVID-19** | | | | |
| Yes | 5 | 1302255 | 774292 | 0.52 |
| No | 1210 | 251052958 | 996138 | 99.48 |

Source: Coronavirus Attitudes and Behaviors Survey, 2020.

have a bachelor's or professional degree are underrepresented (30 percent in the sample versus 36 percent nationally).

Table 2 shows the sample distribution on health factors including diagnosed conditions that increase the risk of severe illness resulting from COVID-19 infection, having experienced COVID-19 symptoms, having been tested for COVID-19, and testing positive for COVID-19. The majority of respondents reported a diagnosis for one or more health conditions that are risk factors for severe illness from COVID-19 (52 percent). The most common conditions were high blood pressure (30 percent), depression or anxiety (25 percent), followed by respiratory condition (16 percent). Less common conditions were diabetes (9 percent), heart condition (8 percent), autoimmune disorder (5 percent), and kidney disease (2 percent).

Approximately 1 in 10 respondents reported having experienced COVID-19 symptoms. As of May 2020, 1 in 20 respondents had been tested for COVID-19, and less than 1 percent (0.5 percent) tested positive for the virus. This is comparable to reported national rates of COVID for this period as reported by the CDC [20].

The literature review indicates that those who are most vulnerable (lower-income households, households with children, and households headed by racial and ethnic minorities) are disproportionately experiencing negative financial impact from the COVID-19 pandemic. Table 3 shows the distribution of the analysis sample on several factors that describe household financial status. The first measure indicates how well the respondent is managing household finances, with 1 in 5 respondents reporting that they are just getting by, and 11 percent reporting that they are having difficulty getting by. The survey also asks about strategies that would

**Table 3. Financial factors.**

|  | Sample Size | Weighted Frequency | Standard Error | Percentage |
|---|---|---|---|---|
| **Total** | 1222 | 253815197 | 3324800 | 100.00 |
| **Financial Management** |  |  |  |  |
| Difficult to get by | 89 | 28081949 | 3147397 | 11.13 |
| Just getting by | 187 | 50411041 | 4252633 | 19.98 |
| Doing okay | 534 | 108056047 | 5148955 | 42.82 |
| Living comfortably | 403 | 65817195 | 4486501 | 26.08 |
| **Strategies for meeting emergency expense** |  |  |  |  |
| Borrow from family or friends | 130 | 43218888 | 3917130 | 21.52 |
| Payday loan | 43 | 12705178 | 2473374 | 6.42 |
| Sell something | 157 | 47411465 | 4496956 | 23.76 |
| Avoid payment | 170 | 53020162 | 3909146 | 28.14 |
| Bank loan | 67 | 20075248 | 3384543 | 10.03 |
| Credit card | 892 | 170161980 | 10048921 | 39.13 |
| Cash | 681 | 140079339 | 4416525 | 65.24 |
| **Financial Vulnerability** |  |  |  |  |
| None | 701 | 114062565 | 4103436 | 44.94 |
| 1 | 224 | 50354533 | 3994529 | 19.84 |
| 2 | 98 | 25009786 | 2741508 | 9.85 |
| 3 | 64 | 19737907 | 3399066 | 7.78 |
| 4 | 61 | 20202822 | 2784622 | 7.96 |
| 5 | 38 | 13843768 | 2589201 | 5.45 |
| 6 to 8 | 36 | 10603816 | 3549674 | 4.18 |
| **Essential Worker** |  |  |  |  |
| Yes | 349 | 71453046 | 3985704 | 29.90 |
| No | 824 | 167545605 | 4141565 | 70.10 |
| **Change in hours worked outside of home** |  |  |  |  |
| Decreased a lot | 515 | 98490133 | 4811691 | 52.38 |
| Decreased somewhat | 84 | 19698965 | 2789492 | 10.48 |
| No change | 234 | 55411580 | 4523352 | 29.47 |
| Increased somewhat | 31 | 7876010 | 2057801 | 4.19 |
| Increased a lot | 22 | 6570374 | 1654239 | 3.49 |

Source: Coronavirus Attitudes and Behaviors Survey, 2020.

be employed if faced with an emergency expense ($400). The majority of the respondents (65 percent) reported that they would use cash to meet an emergency expense, while 2 in 5 respondents indicated that they would use a credit card for this purpose, and 1 in 10 would obtain a bank loan. The remaining strategies reflect limited access to cash or credit. The most common of these is to avoid payment of the obligation (28 percent), followed by selling something (24 percent), or borrowing from family or friends (22 percent). The least common strategy was to obtain a payday loan (6 percent).

In order to capture the financial context in modeling the financial impact of COVID on households, we developed a composite measure of financial vulnerability. Recent research in the area of economics have employed measures of financial vulnerability that capture both financial instability and levels of debt [21, 22]. For the purposes of examining financial vulnerability and disproportionate impact of the COVID-19 pandemic, we examine available resources (household income, access to cash or credit in an emergency) and ability to meet

household expenses. This measure is an index variable, where each subject receives points on the index scale under the following conditions: 1) lower gross annual household income (2 points for annual household income less than $25,000, 1 point for annual household income greater than $25,000 and less than $50,000; 2) difficulty meeting household expenses (2 points for difficulty getting by, 1 point for just getting by); and 3) one point for each strategy for meeting an emergency expense that indicates limited cash reserves or access to credit (borrowing from family or friends, obtaining a payday loan, selling something, or avoiding payment). The score on this index ranges from 0 to 8. Just over one-half of the analysis sample (55 percent) exhibit at least one of these markers for financial vulnerability, with 1 in 3 respondents having a score of 2 or more on this measure.

Household financial status is also examined in regards to employment factors. Table 3 shows that 2 in 3 respondents experienced some reduction in hours worked, while 30 percent were essential workers.

## Outcomes

The impact of the COVID-19 pandemic on household finances is examined on five factors. The first two are indicators of change in household income and typical weekly spending compared to the prior month. Table 4 shows that close to one-half of respondents (44 percent) experienced a decrease in household income, and approximately 2 out of 3 respondents (64 percent) reported a decrease in typical weekly spending.

Respondents were also given the open text prompt, "Please describe in your own words how you are managing financially during the pandemic." Responses were uploaded to NVivo

**Table 4. Financial impact outcomes.**

|  | Sample Size | Weighted Frequency | Standard Error | Percentage |
|---|---|---|---|---|
| Total | 1222 | 253815197 | 3324800 | 100.00 |
| **Change in income** |  |  |  |  |
| Decreased a lot | 162 | 44782593 | 4455113 | 17.85 |
| Decreased somewhat | 287 | 65555383 | 4883951 | 26.13 |
| No change | 673 | 122049838 | 4931792 | 48.66 |
| Increased somewhat | 74 | 16471543 | 2218313 | 6.57 |
| Increased a lot | 11 | 1987607 | 799963 | 0.79 |
| **Change in spending** |  |  |  |  |
| Decreased a lot | 205 | 46340516 | 4056116 | 18.37 |
| Decreased somewhat | 614 | 115849395 | 5050220 | 45.93 |
| No change | 237 | 54839253 | 4561537 | 21.74 |
| Increased somewhat | 142 | 31514994 | 2918049 | 12.49 |
| Increased a lot | 14 | 3707591 | 1110816 | 1.47 |
| **Negative impact** |  |  |  |  |
| Yes | 218 | 44416914 | 3445397 | 17.50 |
| No | 1004 | 209398283 | 3445397 | 82.50 |
| **Neutral impact** |  |  |  |  |
| Yes | 532 | 96315996 | 4231512 | 37.95 |
| No | 690 | 157499201 | 4231512 | 62.05 |
| **Positive impact** |  |  |  |  |
| Yes | 118 | 19416005 | 2428251 | 7.65 |
| No | 1104 | 234399192 | 2428251 | 92.35 |

Source: Coronavirus Attitudes and Behaviors Survey, 2020.

Version 11 for analysis. Inductive coding was used to classify responses. Three broad classifications emerged: negative financial impact, neutral financial impact, and positive financial impact. Subcodes were inductively derived within each of these classifications. Approximately 1 in 5 respondents (18 percent) described a negative financial impact, including increased cost of food, loss of income, and being unable to fully pay household bills. A neutral impact was described by 38 percent of respondents, including those who received government assistance, those who retired, and those who experienced no change in household finances. Only 8 percent indicated that they had experienced a positive financial impact, including reduced expenses, reduced spending, and a stable income.

## Results

The first step of the analysis is an examination of demographic (including age, gender, race, household size, and education), health (risk factors for experiencing severe COVID-19 outcomes), and employment factors (essential worker status and reduction in work hours) that predict financial vulnerability. Table 5 shows the results of a multivariate linear regression analysis, predicting financial vulnerability. Each of these factors was tested first in bivariate models, and all significant bivariate predictors were entered into the multivariate model. Non-significant factors (p-values less than 0.05) in the multivariate model were then removed using a backwards selection process. The model results indicate that Hispanic respondents, those with less than a Bachelor's degree, and those with one or more health risk factors were more vulnerable financially. White respondents and those who were married were less financially vulnerable.

The second step in the analysis was to examine the influence of financial vulnerability, demographics, health factors and employment factors on the impact of COVID on household finances. Financial impact is examined in terms of loss of income; decrease in spending, reported negative impact (increased cost of food, loss of income, and being unable to fully pay household bills); reported neutral impact (receiving government assistance, retiring, and experiencing no change in household finances); and reported positive impact (reduced expenses, reduced spending, or stable income).

Table 6 displays the results of the multivariate logistic regression analyses, predicting the impact of the COVID-19 pandemic on household finances. For the purposes of this analysis, financial vulnerability is reduced to 2 levels, allowing a comparison of those with 2 or more vulnerability factors to those with 1 or no vulnerability factors. Respondents with only 1 vulnerability factor were included in the reference group to provide greater sensitivity in the

**Table 5. Predictors of financial vulnerability–demographic, health, and employment factors (multivariate linear regression).**

| Predictor | Regression Coefficient | Standard Error | p-value |
|---|---|---|---|
| **Race/Ethnicity** | | | |
| Hispanic | 0.66 | 0.25 | 0.0095 |
| White | -0.65 | 0.21 | 0.0024 |
| **Marital Status** | | | |
| Married | -0.92 | 0.15 | <0.0001 |
| **Education** | | | |
| Less than Bachelor's degree | 0.95 | 0.11 | <0.0001 |
| **Health Risk Factors** | | | |
| 1 or more | 0.48 | 0.15 | 0.0013 |

Source: Coronavirus Attitudes and Behaviors Survey, 2020.

**Table 6. Predictors of COVID-19 impact on household finances–financial vulnerability, demographic, health, and employment factors (multivariate logistic regression).**

| Predictor | Loss of Income | Reduced Spending | Negative Impact | Neutral Impact | Positive Impact |
|---|---|---|---|---|---|
| **Financial Vulnerability** | | | | | |
| 2 or more vulnerability factors | 6.33*** | ns | 6.67*** | 0.10*** | 0.04* |
| **Race/Ethnicity** | | | | | |
| Hispanic | 1.82* | ns | ns | ns | 2.71* |
| **Education** | | | | | |
| At least a Bachelor's degree | ns | 1.64** | ns | ns | ns |
| **Employment** | | | | | |
| Non-essential worker | ns | 1.52* | ns | ns | ns |

Source: Coronavirus Attitudes and Behaviors Survey, 2020.

NOTE: Figures in cells represent the odds ratios.

† = p<0.10

* = p<05

** = p<0.01

*** = p<0.001.

indicator variable, as those with only 1 vulnerability factor (e.g., lower household income), would also have some protective factors (e.g., able to manage household finances sufficiently, while also having access to credit or cash in an emergency).

Individuals who were financially vulnerable were 6 times more likely to experience reduced income during the pandemic, while Hispanics were almost twice as likely to have a loss of income. Individuals with less than a college degree and essential workers were less likely to have reduced spending during the pandemic (by 39 percent and 31 percent, respectively). Those who were financially vulnerable were almost 7 times more likely to experience a negative financial impact from the pandemic, while they were less likely to experience a neutral or positive financial impact (by 90 percent and 96 percent, respectively). For perspective, note that those who were not financially vulnerable were 10 times more likely to experience a neutral financial impact, and 25 times more likely to experience a positive financial impact of the coronavirus pandemic.

## Discussion

The novel coronavirus disease pandemic (COVID-19) that began in Spring 2020 has had a substantial impact on the ability of individuals to effectively manage household finances. Researchers are only beginning to examine the nature and extent of this impact, while identifying the population segments that are experiencing the most severe impact and informing policy for providing the necessary supports for those who are struggling most. Current studies of the financial impact of the pandemic on individuals and households are based on aggregate public data, surveys that are not nationally representative, and cross-sectional surveys. This study employs a nationally-representative household survey to examine how the pandemic was affecting household finances in the earliest months of the pandemic. Although our study is descriptive rather than causal, and results should be interpreted with caution, they nevertheless help to illuminate the conditions facing American households as of May 2020. While identifying the subpopulations who are experiencing the most severe hardships (and those who appear more resilient), we also describe patterns of adaption to financial hardship, such as reductions in household spending.

As anticipated, and consistent with recent research findings, we find that those who are financially vulnerable (lowest incomes, limited access to cash and credit, and struggling to meet ordinary expenses) are most likely to experience negative financial impact from the coronavirus pandemic. The financially vulnerable are 6 times more likely to experience loss of income and nearly 7 times more likely to be negatively impacted by the pandemic. Those who are not financially vulnerable are 10 times more likely to experience neutral impact and 25 times more likely to experience positive impact during the pandemic. Those who are most likely to be financially vulnerable include Hispanics, those who are non-White, unmarried, and those who have less than a Bachelor's degree. This group also includes individuals who have health conditions that put them at greater risk of severe illness or death if they contracted the virus.

In terms of race, we find that the increased likelihood of experiencing a negative impact for non-Whites is explained by a greater likelihood of this group being financially vulnerable. However, Hispanic ethnicity directly predicts loss of income. After controlling for financial vulnerability, Hispanics still have a greater likelihood of experiencing a loss of income during the pandemic by 82 percent. Interestingly, Hispanics also have a greater likelihood of experiencing a positive impact from the pandemic (nearly 3 times more likely), compared to non-Hispanics, when controlling for financial vulnerability. While on the surface, these findings may appear contradictory, it is possible that, even in the presence of reduced household income, some individuals may have been able to reduce expenses and spending sufficiently to have seen an overall improvement in their finances during the COVID-19 pandemic.

Reduced spending during the pandemic was the only outcome that did not vary by financial vulnerability. This outcome was predicted only by education level and essential worker status. Those with at least a Bachelor's degree were 64 percent more likely to reduce their spending during the pandemic, while non-essential workers were 50 percent more likely to reduce spending. This may be due to the availability of discretionary income on the part of college-educated respondents and those who continued to work as essential workers during the pandemic.

## Conclusions

While government-funded assistance was broadly available during the pandemic in the form of stimulus checks and expanded unemployment benefits, these measures were not sufficient to buffer the negative financial impact of the pandemic for those who are most vulnerable financially. This group includes racial and ethnic minorities, those who are unmarried, those with lower levels of education, and those with health conditions that put them at higher risk of severe outcomes from the coronavirus. This population, already struggling to provide basic necessities for themselves and their families prior to the pandemic, have been faced with reduced opportunities for earning wages and lack sufficient public support to buffer the impact of this public health crisis. Governmental efforts to counter the financial impact of the pandemic should be targeted to these most vulnerable individuals. Future research should examine the longer-term impact of the COVID-19 pandemic on household finances, in order to observe the extent to which individuals are able to adapt to the changing economy, or the extent to which financial hardship may persist or increase as the pandemic and associated lockdowns are eased or extended.

## Supporting information

**S1 File. Stanford university school of medicine coronavirus attitudes and behaviors survey.** Survey Questionnaire.
(DOCX)

## Acknowledgments

The authors thank the Stanford University School of Medicine team, led by Dr. M. Kate Bundorf, for their contributions to the development and fielding of this study.

## Author Contributions

**Conceptualization:** Carol Bruce, Maeve E. Gearing, Jill DeMatteis, Kerry Levin, Timothy Mulcahy, Jocelyn Newsome, Jonathan Wivagg.

**Data curation:** Jill DeMatteis, Jonathan Wivagg.

**Formal analysis:** Carol Bruce, Maeve E. Gearing, Jill DeMatteis.

**Investigation:** Jill DeMatteis, Kerry Levin, Timothy Mulcahy, Jocelyn Newsome, Jonathan Wivagg.

**Methodology:** Carol Bruce, Maeve E. Gearing, Jill DeMatteis, Kerry Levin, Timothy Mulcahy, Jocelyn Newsome, Jonathan Wivagg.

**Project administration:** Maeve E. Gearing, Timothy Mulcahy, Jonathan Wivagg.

**Software:** Carol Bruce.

**Supervision:** Maeve E. Gearing, Timothy Mulcahy, Jonathan Wivagg.

**Validation:** Jill DeMatteis.

**Visualization:** Carol Bruce.

**Writing – original draft:** Carol Bruce.

**Writing – review & editing:** Maeve E. Gearing.

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
