## [Decision Letter · Decision Letter 0]

22 Jul 2021

PONE-D-21-15762

Financial vulnerability and the impact of COVID-19 on American households

PLOS ONE

Dear Dr. Gearing,

Thank you for submitting your manuscript to PLOS ONE. After careful consideration, we feel that it has merit but does not fully meet PLOS ONE’s publication criteria as it currently stands. Therefore, we invite you to submit a revised version of the manuscript that addresses the points raised during the review process.

We look forward to receiving your revised manuscript.

Kind regards,

Ali B. Mahmoud, Ph.D.

Academic Editor

PLOS ONE

Journal Requirements:

2. Please provide additional details regarding participant consent. In the ethics statement in the Methods and online submission information, please ensure that you have specified whether consent was informed.

3. Please include additional information regarding the survey or questionnaire used in the study and ensure that you have provided sufficient details that others could replicate the analyses. For instance, if you developed the survey or questionnaire as part of this study and it is not under a copyright more restrictive than CC-BY, please include a copy, in both the original language and English, as Supporting Information. If the questionnaire is published, please provide a citation to the (1) questionnaire and/or (2) original publication associated with the questionnaire.

We note that one or more of the authors are employed by a commercial company: Westat.

4.1. Please provide an amended Funding Statement declaring this commercial affiliation, as well as a statement regarding the Role of Funders in your study. If the funding organization did not play a role in the study design, data collection and analysis, decision to publish, or preparation of the manuscript and only provided financial support in the form of authors' salaries and/or research materials, please review your statements relating to the author contributions, and ensure you have specifically and accurately indicated the role(s) that these authors had in your study. You can update author roles in the Author Contributions section of the online submission form.

4.2. Please also provide an updated Competing Interests Statement declaring this commercial affiliation along with any other relevant declarations relating to employment, consultancy, patents, products in development, or marketed products, etc.  

Reviewers' comments:

Reviewer's Responses to Questions

**Comments to the Author**

1. Is the manuscript technically sound, and do the data support the conclusions?

Reviewer #1: Partly

Reviewer #2: Yes

2. Has the statistical analysis been performed appropriately and rigorously? 

Reviewer #1: I Don't Know

Reviewer #2: Yes

3. Have the authors made all data underlying the findings in their manuscript fully available?

Reviewer #1: Yes

Reviewer #2: No

4. Is the manuscript presented in an intelligible fashion and written in standard English?

Reviewer #1: Yes

Reviewer #2: Yes

5. Review Comments to the Author

Reviewer #1: See attached PDF.

Reviewer #2: The article examines the impact of COVID-19 on financial status and how this impact varies by demographic characteristics, the presence of health risk factors, and financial factors. The survey suggests that financial impacts are different especially to the financial vulnerable group, which experience financial difficulty even during non-pandemic situation. This paper addresses a topic of keen interest to several readers of the PLOS ONE. The research design in this paper is sufficient, appropriate and clearly justified. The empirical results appear to be competently implemented and the results have been clearly explained. The present draft needs minor modification and clarification as follow:

1) Please check the write up line 145. The sentence is as follow; …2 in 5 respondents reporting that they are just getting by…In Table 3, the percentage of “Just Getting By” is 19.98% but the 2 in 5 respondents show 40%.

2) Spelling error on line 255 “…wo are struggling”

3) Line 279 to 283 states that “After controlling for financial vulnerability, Hispanics still have a greater likelihood of experiencing a loss of income during the pandemic by 82 percent. Interestingly, Hispanics also have a greater likelihood of experiencing a positive impact from the pandemic (nearly 3 times more likely), compared to non-Hispanics, when controlling for financial vulnerability.” This statement seems contradict. Hispanics group appear to loss of income but experience positive impact from the pandemic. The statements need more clarification.

4) Line 287 to 289. “Those with at least a Bachelor’s degree were 64 percent more likely to reduce their spending during the pandemic, while non-essential workers were 50 percent more likely to reduce spending.” Please check the percentage accordingly in Table 6.

5) The paper could point out the areas for additional research work.

6. PLOS authors have the option to publish the peer review history of their article (what does this mean?). If published, this will include your full peer review and any attached files.

Reviewer #1: No

Reviewer #2: No

---

## [Author Response · Author response to Decision Letter 0]

30 Sep 2021

Reviewer #1

The reviewer’s point that our work does not include several previously-validated measures of financial vulnerability is well-taken. The paper is, generally, descriptive and a-historical. This is the result of the study design itself. We did not set out to examine financial vulnerability to COVID. Rather, we wanted to capture a snapshot of how Americans were responding to COVID in May 2020, along multiple axes of interest, including behavior, finances, and emotional well-being. As such, questions about finances were only one part of the questionnaire and included validated measures asking about changes in income and ability to cope with an unexpected expense. Our interest in financial vulnerability emerged upon analysis of the survey results, when we noted large disparities in the ability to cope financially with the pandemic reported by survey respondents. Hindsight being 20/20, we certainly wish we had included other measures of financial vulnerability, but we hope that our descriptive results may yet provide a fuller understanding of how some Americans experienced financial stress early in the pandemic. We have also added additional citations to the literature on financial vulnerability to further contextualize our results.

Page 3, lines 72-74: Corrected to “lacked” emergency funds

Page 5, lines 108-109: Specified that this referred to gross annual income

Page 8, table 2: Added footnote to clarify that the national incidence of COVID-19 among the U.S. adult population was 0.7, which falls within the 95% confidence interval of our sample.

Page 11, table 4: This was an error made by the corresponding author and has been fixed.

Page 14, table 6: Added text clarifying analysis and a footnote describing index more clearly

Page 16, lines 277-283: Added additional text to contextualize results and their implications

Reviewer #2

Line 145: Corrected

Line 255: Corrected spelling error

Lines 279-283: Added text to clarify results more

Lines 287-289: The table the text referred to was showing the incorrect figures; this was corrected.

Also added additional areas for research

---

## [Decision Letter · Decision Letter 1]

8 Nov 2021

PONE-D-21-15762R1Financial vulnerability and the impact of COVID-19 on American householdsPLOS ONE

Dear Dr. Gearing,

Thank you for submitting your manuscript to PLOS ONE. After careful consideration, we feel that it has merit but does not fully meet PLOS ONE’s publication criteria as it currently stands. Therefore, we invite you to submit a revised version of the manuscript that addresses the points raised during the review process.

We look forward to receiving your revised manuscript.

Kind regards,

Ali B. Mahmoud, Ph.D.

Academic Editor

PLOS ONE

Journal Requirements:

Reviewers' comments:

Reviewer's Responses to Questions

**Comments to the Author**

1. If the authors have adequately addressed your comments raised in a previous round of review and you feel that this manuscript is now acceptable for publication, you may indicate that here to bypass the “Comments to the Author” section, enter your conflict of interest statement in the “Confidential to Editor” section, and submit your "Accept" recommendation.

Reviewer #1: (No Response)

Reviewer #2: All comments have been addressed

Reviewer #3: All comments have been addressed

2. Is the manuscript technically sound, and do the data support the conclusions?

Reviewer #1: Yes

Reviewer #2: Yes

Reviewer #3: Yes

3. Has the statistical analysis been performed appropriately and rigorously? 

Reviewer #1: Yes

Reviewer #2: Yes

Reviewer #3: Yes

4. Have the authors made all data underlying the findings in their manuscript fully available?

Reviewer #1: Yes

Reviewer #2: Yes

Reviewer #3: Yes

5. Is the manuscript presented in an intelligible fashion and written in standard English?

Reviewer #1: Yes

Reviewer #2: Yes

Reviewer #3: (No Response)

6. Review Comments to the Author

Reviewer #1: Thanks for trying to address my comments on the previous version of the paper. I note that the paper by Anderloni et al. (2012) has been published, but is mentioned as a "departmental working paper" in the reference list.

Please correct to: Anderloni, L., Bacchiocchi, E., & Vandone, D. (2012). Household financial vulnerability: An empirical analysis. Research in Economics, 66(3), 284-296.

I also notice that the reference to the paper by Hoffmann and McNair (2018) contains typos in the author names. In particular "Hoffman" should be "Hoffmann". Please correct this as well.

Reviewer #2: (No Response)

Reviewer #3: Given the very poor response rate, despite the weighting, the analysis can best be described as descriptive. The statistical tools used are routine. The authors should at least note a serious limitation in the discussion as to the low response rate and the fact that the results should be interpreted with precaution.

7. PLOS authors have the option to publish the peer review history of their article (what does this mean?). If published, this will include your full peer review and any attached files.

Reviewer #1: No

Reviewer #2: No

Reviewer #3: No

---

## [Author Response · Author response to Decision Letter 1]

20 Dec 2021

Reviewer #1

Thank you for the corrections on the references. These have been addressed.

Reviewer #3

Thank you for your comments. We agree that this is a descriptive analysis; although the response rate is in line with similar nationally-representative online surveys, we do want to be cautious in interpretation, as we cannot determine causality. We have added caveats and cautions about interpretation to our discussion.

---

## [Decision Letter · Decision Letter 2]

21 Dec 2021

Financial vulnerability and the impact of COVID-19 on American households

PONE-D-21-15762R2

Dear Dr. Gearing,

We’re pleased to inform you that your manuscript has been judged scientifically suitable for publication and will be formally accepted for publication once it meets all outstanding technical requirements.

Kind regards,

Ali B. Mahmoud, Ph.D.

Academic Editor

PLOS ONE

Additional Editor Comments (optional):

Reviewers' comments:

Reviewer's Responses to Questions

**Comments to the Author**

1. If the authors have adequately addressed your comments raised in a previous round of review and you feel that this manuscript is now acceptable for publication, you may indicate that here to bypass the “Comments to the Author” section, enter your conflict of interest statement in the “Confidential to Editor” section, and submit your "Accept" recommendation.

Reviewer #3: All comments have been addressed

2. Is the manuscript technically sound, and do the data support the conclusions?

Reviewer #3: (No Response)

3. Has the statistical analysis been performed appropriately and rigorously? 

Reviewer #3: (No Response)

4. Have the authors made all data underlying the findings in their manuscript fully available?

Reviewer #3: (No Response)

5. Is the manuscript presented in an intelligible fashion and written in standard English?

Reviewer #3: (No Response)

6. Review Comments to the Author

Reviewer #3: (No Response)

7. PLOS authors have the option to publish the peer review history of their article (what does this mean?). If published, this will include your full peer review and any attached files.

Reviewer #3: No

---

## [Editor Report · Acceptance letter]

31 Dec 2021

PONE-D-21-15762R2 

Financial vulnerability and the impact of COVID-19 on American households 

Dear Dr. Gearing:

I'm pleased to inform you that your manuscript has been deemed suitable for publication in PLOS ONE. Congratulations! Your manuscript is now with our production department. 

Kind regards, 

on behalf of

Dr. Ali B. Mahmoud 

Academic Editor

PLOS ONE